

# Prepollination barriers prevent gene flow between co-occurring bat-pollinated bromeliads in a montane forest

Stephanie Núñez-Hidalgo[1] and Alfredo Cascante-Marín[2]

[1] Escuela de Ciencias Ambientales, Universidad Nacional, Heredia, Costa Rica
[2] Centro de Investigaciones en Biodiversidad y Ecología Tropical (CIBET) & Escuela de Biología, Universidad de Costa Rica, San Pedro de Montes de Oca, San José, Costa Rica

## ABSTRACT

**Background:** Reproductive isolation mechanisms in flowering plants are fundamental to preserving species' evolutionary independence and to enabling the local coexistence of closely related species. These reproductive barriers are expected to contribute to maintaining local diversity of highly diverse plant guilds, such as bromeliads in neotropical ecosystems. We evaluated how strong and effective these barriers are by analyzing different mechanisms that act before and after pollination in a guild of four epiphytic bromeliads from the genus *Werauhia* (Tillandsioideae) pollinated by bats in a Costa Rican montane forest.

**Methods:** We employed several reproductive isolation indices proposed in the literature to estimate the effect of flowering phenology, floral morphology, interspecific compatibility, production, and viability of hybrid seeds as barriers to gene flow between species pairs.

**Results:** The overall reproductive isolation between species was complete or nearly so. We found that temporal isolation due to different flowering schedules between species significantly contributed to preventing interspecific gene flow. However, flowering data from four reproductive seasons showed interannual variation in the intensity of this temporal barrier due to fluctuations in the species' blooming patterns. For species with overlapping flowering, mechanical isolation caused by differences in flower size and position of reproductive organs was significant, and such differences in flower architecture are thought to influence pollen deposition on different areas of the pollinator's body. Postpollination barriers showed varying intensity, from full to partial interspecific incompatibility. When hybrid progeny was produced, the number of seeds and their germination capacity were lower compared to progeny from intraspecific crosses.

**Conclusions:** Overall, prepollination mechanisms (phenology and floral design) were of great importance to eliminate pollen transfer between species and, when present, postpollination barriers had a redundant effect. Our results contradict previous reports that suggested a weak effect of premating barriers among bromeliad species. Additional studies involving other pollination guilds are required to gain a better understanding of the prevalence of different reproductive isolation mechanisms in the highly diverse Bromeliaceae family.

Corresponding author
Alfredo Cascante-Marín,
alfredo.cascante@ucr.ac.cr

## INTRODUCTION

Reproductive isolation is a fundamental driver of plant diversity (*Baack et al., 2015*) by preventing reproductive interference and facilitating the simultaneous coexistence of closely related species (*Schemske, 2010*). Flowering plant species that share the same habitat frequently employ a variety of reproductive isolation mechanisms to prevent interspecific pollen transfer and hybridization (*Coyne & Orr, 2004*; *Lowry et al., 2008*; *Widmer, Lexer & Cozzolino, 2009*), which may result in the loss of gametes and the formation of nonviable hybrids (*Campbell & Aldridge, 2006*; *Moreira-Hernández & Muchhala, 2019*).

Reproductive isolation mechanisms restrict gene flow between species and consist of floral differences of a morphological, ethological, physiological, or genetic nature and can be classified into two types according to whether they occur before or after pollination, also referred to as pre- and postmating barriers, respectively (*Levin, 1971*; *Baack et al., 2015*; *Campbell & Aldridge, 2006*). The degree of pre- and postpollination isolation can vary among species and may be influenced by the pollination system (*Cozzolino, D'Emerico & Widmer, 2004*; *Cozzolino & Scopece, 2008*). Generally, the efficiency of reproductive isolation mechanisms is complemented sequentially at each stage, that is, a reproductive barrier prevents gene flow that was not eliminated by previous barriers (*Widmer, Lexer & Cozzolino, 2009*).

Based on evidence from the past 20 years (*Lowry et al., 2008*; *Baack et al., 2015*; *Christie, Fraser & Lowry, 2022*), prepollination barriers appear to be significantly more effective than postpollination barriers, with floral isolation mechanisms being more robust. Most published data on reproductive isolation barriers (82%) are from temperate plant groups that mainly include herbaceous and perennial species from the Orobanchaceae and Orchidaceae families pollinated by insects (*Schiestl & Schlüter, 2009*; *Christie, Fraser & Lowry, 2022*). Reproductive barriers operating in neotropical plant lineages with specialized pollination systems are poorly understood and have only been the subject of recent investigations (*e.g.*, *Kay, 2006*; *Cuevas, Espino & Marques, 2018*; *Ramírez-Aguirre et al., 2019*; *Arida et al., 2021*; *Albuquerque-Lima, Lopes & Machado, 2024*). Understanding the species coexistence and maintenance of plant diversity in highly diverse tropical ecosystems requires further comparative studies of reproductive isolation mechanisms.

The Bromeliaceae family is an example of a highly diverse lineage (ca. 3,600 species) that is almost exclusive to the American Continent (except for one species from West Africa) (*Benzing, 2000*). Bromeliad diversity is concentrated in four areas: the Atlantic Forest in eastern Brazil, the Andean slopes, Central America, and the Guiana Highlands (*Zizka et al., 2020*). The great morphological diversity in bromeliads is partially ascribed to hybridization processes that have also contributed to speciation (*Gardner, 1984*; *Schulte et al., 2010*; *Goetze et al., 2017*). However, even in the presence of potential hybridization, the maintenance of high regional and local diversity in bromeliads implies the existence of reproductive isolation mechanisms.

To understand how reproductive coexistence operates in co-occurring congeneric species of bromeliads and how local plant diversity is maintained, we estimated the strength and relative contribution of several pre- and postpollination reproductive barriers in a group of four sympatric *Werauhia* species in a montane forest in Costa Rica. The mountains of southern Central America extending from Costa Rica to western Panama represent the radiation center of the genus *Werauhia* J. R. Grant from subfamily Tillandsioideae (*Grant, 1995*). This group of epiphytic and tank-forming bromeliads consists of approximately 100 species (*Gouda & Butcher*) and is distinguished by a combination of nocturnal anthesis, inconspicuous floral coloration (white, cream or greenish), petals with dactyloid-shaped basal appendages with divided apex, and cupular-shaped stigmas without papillae (*Grant, 1995*). *Werauhia* has been retrieved as monophyletic in molecular investigations (*Barfuss et al., 2005*) and appears to have a relatively recent diversification history (about 5 million years) (*Givnish et al., 2011*). Understanding the ecological factors that modulate the species' reproductive coexistence may help elucidate the mechanisms driving their diversification.

Bromeliads are plants of ornamental interest that have been used to develop and cultivate artificial hybrids (*Negrelle, Anacleto & Mitchell, 2012*). This suggests that postpollination mechanisms such as interspecific genetic incompatibility or incongruity (*sensu Knox, Williams & Dumas, 1986*) might not represent an important reproductive barrier in the family. Nonetheless, the rarity of naturally occurring hybrids (*Smith & Downs, 1974*; *Gardner, 1984*; *Benzing, 2000*; *Souza et al., 2017*; *Neri, Wendt & Palma-Silva, 2018*) instead suggests that prepollination barriers could be more effective. However, some authors have advocated the contrary view that bromeliads have inadequate prepollination barriers (*Wendt et al., 2008*; *Matallana et al., 2016*).

This study evaluated four mechanisms of reproductive isolation in sympatry which consider prepollination barriers: (i) temporal barriers related to population floral phenology, (ii) mechanical floral barriers associated with flower size and position of reproductive organs, and postpollination barriers: (iii) prezygotic barriers related to interspecific incompatibility or incongruity, and (iv) postzygotic barriers related to the production and viability of hybrid seeds. By using a series of indirect methods or reproductive isolation indices (RI) described by *Sobel & Chen (2014)*, we estimated how much gene flow is reduced by each reproductive barrier and quantified the relative contribution of pre- and postpollination barriers to total reproductive isolation between species pairs. This study is a partial fulfillment of the requirements for a Master Degree of the first author at the graduated program (Sistema de Estudios de Posgrado) from the University of Costa Rica.

## MATERIALS AND METHODS

### Study site

The bromeliads studied were located at a montane forest ecosystem in the Central Valley of Costa Rica (9°52′–9°54′N and 83°57′–84°00′W). La Carpintera Protective Zone is a small mountain formation between 1,500 and 1,850 m asl with nearly 2,400 hectares in size (35% primary forest and 57% old secondary forest) (*Sánchez-González, Duran & Vega,*

*2008*). The study was conducted on the property of Iztarú Field School from the Association of Guides and Scouts of Costa Rica and under permission granted by the former station administrator, Mr. Minor Serrano. The zone receives an average annual rainfall of 1,839.2 mm, with a mean temperature of 16.1 °C, and a mild drier season runs from December to April (*Ríos & Cascante-Marín, 2017*); however, the presence of fog is frequent during the night and early morning. The Life Zones System of *Holdridge (1978)* classifies the vegetation as both humid and very humid lower montane forest. The local diversity of vascular epiphytic plants represents nearly one-third of the local flora, and bromeliads contribute with 29 species; represented by the genera *Aechmea* (1 spp.), *Catopsis* (3 spp.), *Guzmania* (3 spp.), *Pitcairnia* (1 spp.), *Racinaea* (2 spp.), *Tillandsia* (11 spp.), *Vriesea* (1 spp.), and *Werauhia* (7 spp.) (*Sánchez-González, Duran & Vega, 2008*).

## Study species

We selected the four most abundant *Werauhia* species based on previous research on the flowering phenology of epiphytic plants at the study site (*Cascante-Marín, Trejos & Alvarado, 2017*): *W. ampla* (L. B. Sm.) J. R. Grant, *W. nephrolepis* (L. B. Sm. & Pittendr.) J. R. Grant, *W. pedicellata* (Mez & Wercklé) J. R. Grant, and *W. subsecunda* (Wittm.) J. R. Grant. These epiphytic species develop small to medium-sized rosettes, simple spiked (*W. ampla* and *W. subsecunda*), or compound inflorescences (*W. nephrolepis* and *W. pedicellata*) (Fig. 1). These bromeliads share the same pollinator at the study site, the nectar-feeding bat *Hylonycteris underwoodi* (Phyllostomidae) and exhibit a highly self-compatible mating system with an autonomous delayed mechanism of selfing (*Núñez-Hidalgo & Cascante-Marín, 2024*). Their geographic distribution mostly encompasses the very humid and cloudy forests between 1,000 and 2,750 m asl on the Talamanca Mountain range in Southern Central America between Costa Rica and Panama (*Morales, 2003*).

## Prepollination mechanisms
### Temporal isolation by floral phenology

We determined the flowering phenology pattern for each species along four non-consecutive reproductive seasons. For this, we established 10 sampling points with a high density of bromeliads at the study site (1,650–1,780 m asl) and conducted biweekly censuses to document the number of plants opening flowers from a sample of 70 to 385 plants per species. The censuses were carried out from October 2018 to July 2019 and December 2020 to July 2021. We identified that period as the main flowering time based on data from a previous study of phenology of epiphyte plants at the study site (*Cascante-Marín, Trejos & Alvarado, 2017*). We incorporated data from the flowering seasons of 2012–2013 and 2014–2015 previously collected by the second author using the same methodology.

Then, we estimated the RI arising from phenological differences between species pairs following the formula $RI_{4S2}$ proposed by *Sobel & Chen (2014)* and included in the Excel spreadsheet template provided in their supplementary material. This index, hereafter

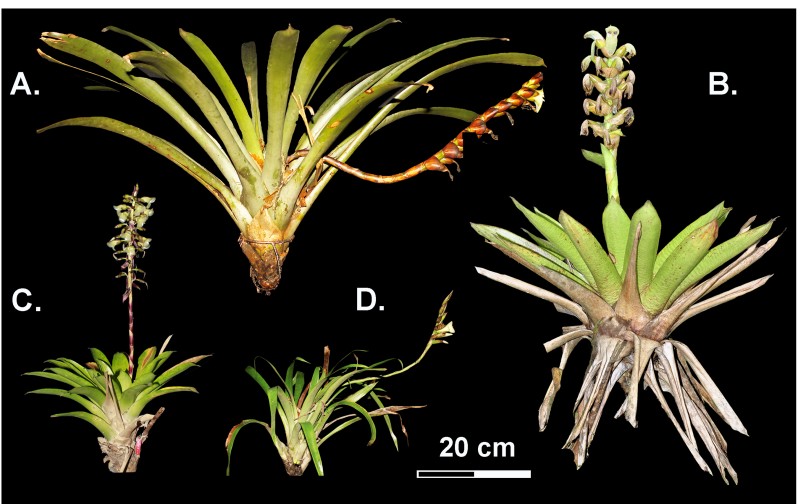

**Figure 1 Reproductive bromeliads from the studied *Werauhia* species (Bromeliaceae) from a montane forest, Costa Rica.** (A) *W. ampla*; (B) *W. nephrolepis*; (C) *W. pedicellata;* (D) *W. subsecunda*. Photo credit: A. Cascante-Marín.   

called $RI_F$, reflects the magnitude of floral asynchrony as a barrier to the formation of hybrids between pairs of species and contemplates shared and unshared flowering days and differences in sample sizes. A $RI_F$ value equal to zero indicates the absence of reproductive barriers, while a value equal to 1 corresponds to complete reproductive isolation. We estimated $RI_F$ for each species pair as a mean value from the four analyzed phenology periods.

### Mechanical isolation by floral morphology

Plants may prevent or reduce interspecific pollen transfer by placing the pollen on different parts of the pollinator's body, and this is achieved through differences in flower size and position of reproductive organs in the corolla (*Dressler, 1981*; *Muchhala, 2008*). Thus, we included two mechanical RI, one for differences in corolla size ($RI_{MS}$) and another for differences in anthers and stigma position ($RI_{MP}$). Consequently, we measured the following floral morphology traits: (i) length of the corolla, (ii) diameter of the corolla aperture, (iii) length of the stamens and (iv) length of the pistil, using a ruler with a precision of one millimeter. The sample consisted of 33 freshly opened flowers from 20 *W. subsecunda* plants, 31 flowers from 15 *W. nephrolepis*, 30 flowers from 15 *W. ampla*, and 30 flowers from 10 *W. pedicellata*.

Since our data on floral morphology did not meet the normality assumption, we used a global PERMANOVA based on a Euclidian distance matrix and 999 permutations to test the significance of the differences in floral traits among species. After a significant result, we conducted pairwise PERMANOVAs with the same conditions to test for differences between species. We used the *RVAideMemoire* package (*Herve, 2023*) and corrected for multiple comparisons by means of a Bonferroni correction. To visualize the differences in flower morphology among species in a multidimensional space, we used the four above mentioned floral traits to perform a principal component analysis (PCA) with the

*FactoMineR* package (*Lê, Josse & Husson, 2008*) in the R software platform (*R Core Team, 2021*). Variables were standardized to unit variance. Upon examining the PCA biplot, we determined that species exhibiting some overlap or unclear separation along either of the PCA dimensions (*i.e.*, possessing similar floral morphologies) have the possibility of gene exchange, indicating a weak morphological barrier. Consequently, we conservatively assigned a value of $RI_{MS} = 0$. If clearly separated in the multidimensional space (*i.e.*, with different floral morphologies), we assigned a $RI_{MS} = 1$, indicating complete isolation due to floral size.

In most *Werauhia* species, the anthers position together forming a hood over the dorsal side of the corolla aperture or, less frequently, the anthers may separate into two groups of three (triplets) and locate on both sides of the corolla aperture with the stigma on either side (*Utley, 1983*). For species pairs sharing the same arrangement of reproductive structures, we assumed no restriction to gene flow and assigned a reproductive isolation index ($RI_{MP}$) value equal to 0 (no isolation), otherwise we assigned a value equal to 1 (complete isolation), since pollen deposition is expected to occur on different parts of the pollinator's body.

## Postpollination mechanisms

To estimate the strength of postpollination barriers, we followed the formula proposed by *Sobel & Chen (2014)*: $RI_{4C} = 1 - 2(H/H + C)$, where H = hetero-specific events (percentage of fruits, seed number or seed germination from interspecific manual crosses), and C = conspecific events (percentage of fruits, number of seeds or seed germination from intraspecific manual crosses). Using the previous formula, we calculated the indices corresponding to: (i) inter-specific incompatibility or $RI_I$ (pre-zygotic barrier), (ii) hybrid seed production or $RI_S$ (post-zygotic barrier) and (iii) seed viability or $RI_V$ (post-zygotic barrier). The RI value indicates the amount of interspecific gene flow at each stage, where $-1$ indicates the presence of inter-specific pollen flow (absence of barriers), 0 indicates random pollen flow, and 1 indicates complete isolation of gene flow between species. Data on intraspecific events (manual cross-pollinations) were obtained from a related work on the breeding systems of the study species at the same site (*Núñez-Hidalgo & Cascante-Marín, 2024*).

### *Interspecific incompatibility*

We conducted controlled interspecific cross-pollinations in a total of 67 plants (18 *W. ampla*, 19 *W. pedicellata*, and 29 *W. subsecunda*) from November 2018 to May 2019. These species showed an overlap in reproductive phenology during the study period and could potentially interbreed. The manipulated plants were kept in a shade house located at the study site at 1,760 m asl. Interspecific manual pollinations were performed reciprocally; thus, a plant was both pollen-donor and pollen-recipient. Before floral anthesis, anthers were carefully removed with a pair of tweezers before dehiscence to avoid contamination of the stigma and stored in paper envelopes until the time of manual pollination in the same night. Pollen in sufficient quantity was applied to receptive stigmas (*i.e.*, with stigmatic fluid present) 1–2 hours after anthesis using a metal spatula. The

respective RI value was calculated with the proportion of fruits developed after each treatment.

### Hybrid progeny and seed viability

To determine the existence of postpollination barriers acting on hybrid progeny formation, we counted the number of seeds per fruit from successful interspecific crosses and compared it to the respective intraspecific crosses. We tested the viability of the hybrid seeds by carrying out a germination test in laboratory conditions. We mixed the seeds from each species-pair cross and distributed a sample of 480 seeds among 12 replicates, each containing 40 seeds. Each replicate was germinated on wet paper towel in Petri dishes. To reduce the incidence of fungal contamination, we once applied a commercial fungicide (Vitabax 40 WP). We determined the number of germinated seeds twice a week, a seed had germinated when the radicle emerged from the seed coat. Once we noticed no seeds germinating, we calculated the cumulative percentage of germination, usually one month after the trial began.

## Contribution of each barrier type and total reproductive isolation

Each reproductive barrier contributes to isolation in proportion to the order in which they occur in the plant's life. As a result, the first barrier to action will reduce gene flow, implying a greater contribution to reproductive isolation (*Coyne & Orr, 1989*; *Ramsey, Bradshaw & Schemske, 2003*). We used the formulas proposed by *Sobel & Chen (2014)* and provided in their supplementary material (evo12362-sup-0003) to estimate the relative and absolute contribution of each type of barrier and total reproductive isolation between each pair of species. We also identified which category of isolation barrier (pre- or postpollination) contributes more to reducing gene flow between species. The calculations were performed using an Excel spreadsheet provided by *Sobel & Chen (2014)* and included here as a Supplemental File.

## RESULTS

### Isolation by floral phenology

The species exhibited an annual pattern of flowering (*sensu Newstrom, Frankie & Baker, 1994*) but the intensity and distribution of flowering peaks varied among species and between years (Fig. 2). The blooming periods were seasonally divided: *W. ampla, W. pedicellata*, and *W. subsecunda* mostly bloomed in the dry season and *W. nephrolepis* flowered separately during the rainy season (Fig. 2). *Werauhia ampla* and *W. subsecunda* showed the longest reproductive periods (5–6 months), both with a pattern of constant intensity or "steady-state" (*sensu Janzen, 1967*) and flowering peaks of relatively low intensity (usually <40% of the observations) (Fig. 2). *W. pedicellata* showed a defined bimodal pattern and a low to moderate overlap with the two previous species. The flowering of *W. nephrolepis* was of short duration with a very marked peak resembling the "cornucopia type" pattern (*sensu Janzen, 1967*) and temporally isolated from the rest (Fig. 2).

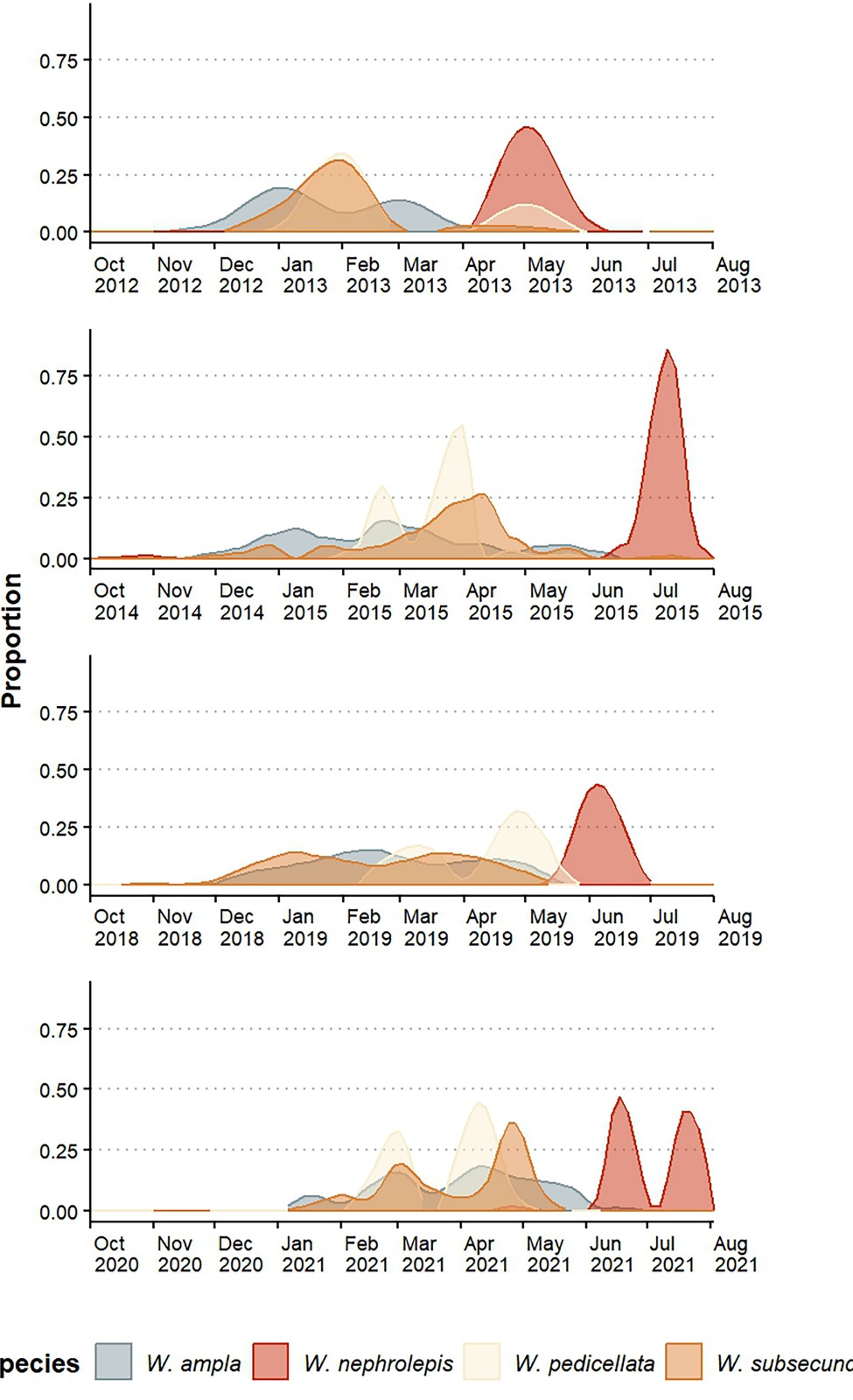

**Species** ▢ *W. ampla*  ▢ *W. nephrolepis*  ▢ *W. pedicellata*  ▢ *W. subsecunda*

**Figure 2 Floral phenology patterns of four epiphytic bromeliads of the genus *Werauhia* in a montane forest, Costa Rica.** Data show the proportion of flowering plants in four reproductive periods.

**Table 1 Estimated values of pre- and postpollination reproductive isolation barriers among four sympatric *Werauhia* (Bromeliaceae) species in a montane forest from Costa Rica.**

| Pollen recipient × pollen donor | | Wa × Wn | | | Wn × Wa | | | Wa × Wp | | | Wp × Wa | | |
|---|---|---|---|---|---|---|---|---|---|---|---|---|---|
| Isolation barrier | RI | Strength | AC | RC | Strength | AC | RC | Strength | AC | RC | Strength | AC | RC |
| Phenology | $RI_F$ | 0.984 | 0.984 | 1 | 0.968 | 0.968 | 1 | 0.497 | 0.497 | 0.497 | 0.467 | 0.467 | 0.467 |
| Floral mechanical—size | $RI_{MS}$ | 0 | 0 | 0 | 0 | 0 | 0 | 1 | 0.503 | 1 | 1 | 0.533 | 0.533 |
| Floral mechanical—position | $RI_{MP}$ | 0 | 0 | 0 | 0 | 0 | 0 | 0 | 0 | 0 | 0 | 0 | 1 |
| Interspecific incompatibility | $RI_I$ | – | – | – | – | – | – | 1 | 0 | 0 | 1 | 0 | 1 |
| **Total isolation** | | | 0.984 | 1 | | 0.968 | 1 | | 1 | 1 | | 1 | 1 |
| | | Wa × Ws | | | Ws × Wa | | | Wn × Ws | | | Ws × Wn | | |
| Phenology | $RI_F$ | 0.128 | 0.128 | 0.128 | 0.258 | 0.258 | 0.258 | 0.967 | 0.967 | 0.967 | 0.983 | 0.983 | 0.983 |
| Floral mechanical—size | $RI_{MS}$ | 1 | 0.872 | 0.872 | 1 | 0.742 | 0.742 | 1 | 0.033 | 0.033 | 1 | 0.017 | 0.017 |
| Floral mechanical—position | $RI_{MP}$ | 1 | 0 | 0 | 1 | 0 | 0 | 1 | 0 | 0 | 1 | 0 | 0 |
| Interspecific incompatibility | $RI_I$ | 1 | 0 | 0 | 0.168 | 0 | 0 | – | – | – | – | – | – |
| Seed production | $RI_S$ | – | – | – | 0.497 | 0 | 0 | – | – | – | – | – | – |
| Seed germination | $RI_V$ | – | – | – | 0.149 | 0 | 0 | – | – | – | – | – | – |
| **Total isolation** | | | 1 | 1 | | 1 | 1 | | 1 | 1 | | 1 | 1 |
| | | Wp × Ws | | | Ws × Wp | | | Wp × Wn | | | Wn × Wp | | |
| Phenology | $RI_F$ | 0.345 | 0.345 | 0.345 | 0.554 | 0.554 | 0.554 | 0.978 | 0.978 | 0.978 | 0.991 | 0.991 | 0.991 |
| Floral mechanical—size | $RI_{MS}$ | 0 | 0 | 0 | 0 | 0 | 0 | 1 | 0.022 | 0.022 | 1 | 0.009 | 0.009 |
| Floral mechanical—position | $RI_{MP}$ | 1 | 0.655 | 0.655 | 1 | 0.446 | 0.446 | 0 | 0 | 0 | 0 | 0 | 0 |
| Interspecific incompatibility | $RI_I$ | 0.097 | 0 | 0 | 0.348 | 0 | 0 | – | – | – | – | – | – |
| Seed production | $RI_S$ | 0.414 | 0 | 0 | 0.316 | 0 | 0 | – | – | – | – | – | – |
| Seed germination | $RI_V$ | 0.487 | 0 | 0 | 0.034 | 0 | 0 | – | – | – | – | – | – |
| **Total isolation** | | | 1 | 1 | | 1 | 1 | | 1 | 1 | | 1 | 1 |

Note:
Values represent the strength of each barrier and their accumulated (AC) and relative contribution (RC) between species pairs. Wa = *W. ampla*, Wn = *W. nephrolepis*, Wp = *W. pedicellata*, and Ws = *W. subsecunda*.

Estimations of temporal isolation between species-pair combinations using the four-year average value of the index were very variable, ranging from 0.128 to 0.991 (Table 1). It was the lowest between *W. ampla* and *W. subsecunda* ($RI_F$ = 0.128 and 0.258), and for all paired comparisons involving *W. nephrolepis*, it indicated strong temporal isolation in both directions ($RI_F \geq 0.97$) (Table 1). There were some variations between years in the strength of this barrier for some species pairs (Table S1).

## Isolation by floral morphology

Differences in floral morphometry among species were statistically significant (PERMANOVA test: $r^2$ = 0.932; DF = 3, 118; F-value = 544.67; *p*-value = 0.001). Similarly, all paired comparisons among species were significant (*p* = 0.006) which indicates substantial variation in flower size. For instance, the length of the corolla, pistil and stamens in *W. ampla* and *W. nephrolepis* were 2–3 times longer compared to *W. subsecunda* and *W. pedicellata* (Fig. 3A). The PCA biplot suggested two distinctive groups based on dimensions of floral parts, large (*W. ampla* and *W. nephrolepis*) *vs.*

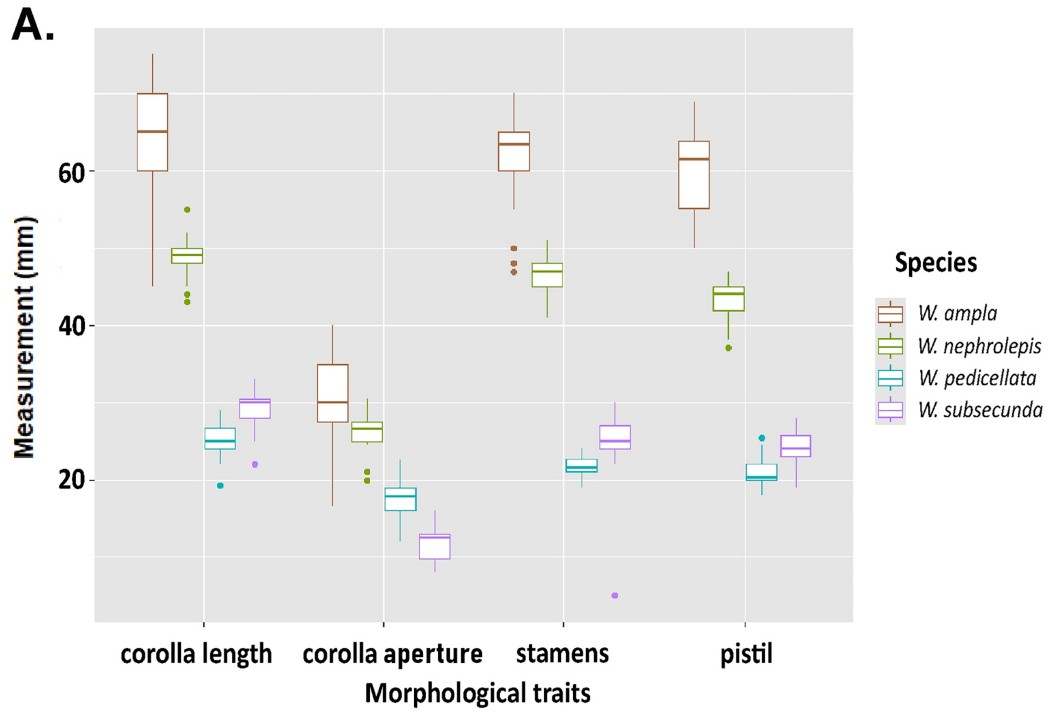

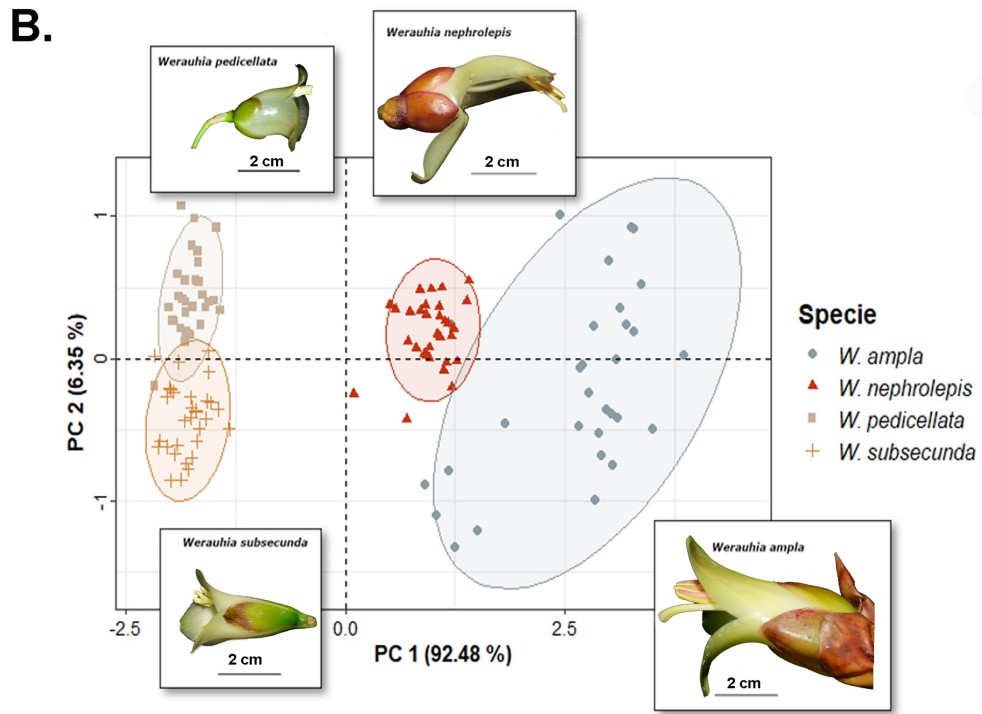

**Figure 3** **Results from the floral morphology analyses based on corolla length, diameter of the corolla aperture, and the lengths of stamens and pistil from four *Werauhia* species in a montane forest, Costa Rica.** (A) Boxplot summarizes data on the quantified floral characteristics for each species. (B) PCA biplot illustrates the first and second principal components; observations are grouped by species (with ellipses indicating the 95% confidence level) and the flower image of each species are superimposed on the graph. Photo-credits: A. Cascante-Marín.

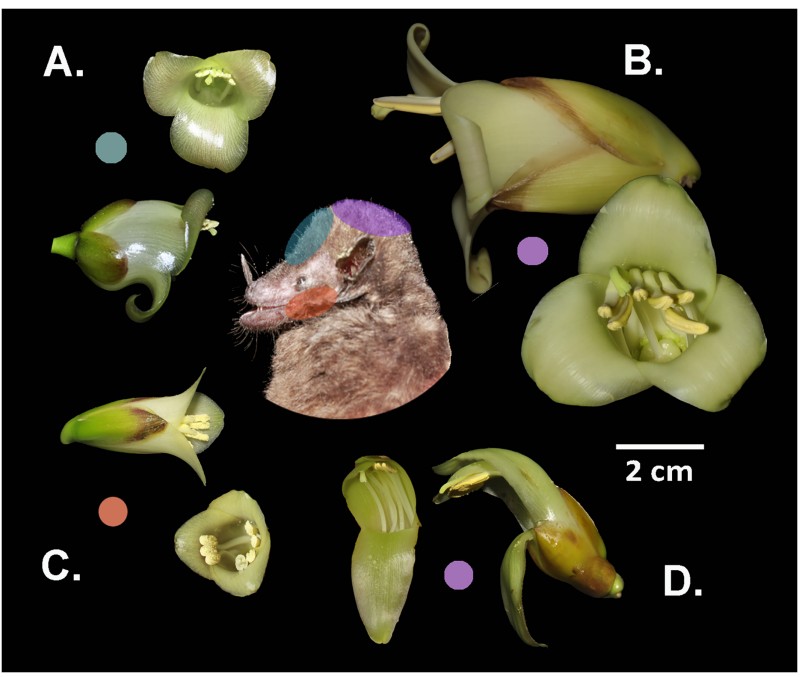

**Figure 4 Differential placement of pollen from the studied *Werauhia* species on the body of their primary bat pollinator, *Hylonycteris underwoodi* (Glossophaginae), in a montane forest from Costa Rica.** Flowers depicted in lateral and frontal perspective. The colored circles represent the distinct sites of pollen deposition from each bromeliad on the bat's body: (A) *W. pedicellata*, (B) *W. ampla*, (C) *W. subsecunda*, (D) *W. nephrolepis*. Photo credits: A. Cascante-Marín & S. Núñez-Hidalgo.

small-flowered (*W. subsecunda* and *W. pedicellata*) species (Fig. 3B). The first component explained most (92.48%) of the variation in the data, with stamens and pistil length showing the highest scores (Table S2). These size differences in reproductive structures were deemed to represent complete isolation ($RI_{MS} = 1$; Table 1), since such morphological dissimilarity precludes effective interspecific pollen transfer between those two species groups.

The species *W. ampla, W. nephrolepis*, and *W. pedicellata* share the same position of stamens and stigma on the dorsal part of the corolla aperture (Fig. 4), suggesting the probability of gene flow ($RI_{MP} = 0$). In *W. subsecunda* the pistil and stamens separate into two triplets that locate in lateral position at the corolla aperture (Fig. 4C). This conformation of the reproductive organs represents a strong barrier to gene flow with respect to the other species, thus paired comparisons involving *W. subsecunda* were assigned a complete reproductive isolation for this aspect of floral morphology ($RI_{MP} = 1$) (Table 1).

## Isolation by interspecific incompatibility

Reciprocal crosses between *W. subsecunda* and *W. pedicellata* showed partial interspecific incompatibility and resulted in fruit percentages of 37% and 47.8% ($RI_I = 0.097$ and $0.348$, respectively; Table 2). Crosses involving *W. ampla* only produced fruits with

**Table 2 Results of reciprocal crosses involving the epiphytic bromeliads *Werauhia ampla*, *W. pedicellata*, and *W. subsecunda* in a montane forest, Costa Rica.**

| Trait | Pollen donor | | |
| --- | --- | --- | --- |
| **Pollen recipient** | *W. ampla* | *W. pedicellata* | *W. subsecunda* |
| **Fruit production (%)** | | | |
| *W. ampla* | **82.4** (17 fls) | 0 | 0 |
| *W. pedicellata* | 0 | **58.1** (31 fls) | 47.8 (23 fls) |
| *W. subsecunda* | 54.5 (33 fls) | 37.0 (27 fls) | **76.5** (17 fls) |
| **Seeds per fruit** | | | |
| *W. ampla* | **2,136 ± 497** (14 frt) | – | – |
| *W. pedicellata* | – | **367 ± 48** (8 frt) | 152 ± 76 (6 frt) |
| *W. subsecunda* | 214 ± 84 (11 frt) | 330 ± 182 (6 frt) | **636 ± 234** (8 frt) |
| **Seed germination (%)*** | | | |
| *W. ampla* | **80.6 ± 15.7** | – | – |
| *W. pedicellata* | – | **92.5 ± 6.7** | 31.9 ± 28.4 |
| *W. subsecunda* | 73.1 ± 12.7 | 92.3 ± 4.9 | **98.8 ± 2.0** |

Notes:
* Data for each species were obtained from 12 replicates of 40 seeds.
Data are mean values ± SD and sample sizes in parentheses from fruit production, number of seeds per fruit, and seed germination capacity. The values on the diagonal (in bold) represent the results from intraspecific outcrosses (taken from *Núñez-Hidalgo & Cascante-Marín, 2024*). Crossings with *W. nephrolepis* are not included because of non-overlapping flowering with the other species.

*W. subsecunda*, when the latter species acted as pollen recipient (54.5%, $RI_I$ = 0.168), and it represented a case of asymmetric incongruity. Because of full incompatibility, the reproductive barriers between *W. ampla* and *W. pedicellata* were complete ($RI_I$ = 1).

### Isolation by hybrid progeny unviability

When compared to the respective intraspecific crossings (Table 2), the number of hybrid seeds per fruit from reciprocal crosses between *W. subsecunda* and *W. pedicellata* resulted in a reduction of 48% and 66.4%, respectively. This represented a relatively low to moderate isolation barrier between the two species ($RI_S$ = 0.316 and 0.414, respectively). When *W. subsecunda* (the pollen recipient) crossed with *W. ampla*, fruits produced about a third as many hybrid seeds as fruits from intraspecific crosses of the same species (214 *vs.* 636 seeds, respectively; Table 2). This represented a moderate barrier to reproduction ($RI_S$ = 0.497).

The germination capacity of hybrid seeds from *W. subsecunda* sired with pollen from *W. pedicellata* was high (92.3%; Table 2). This resulted in a nearly absent isolation barrier ($RI_V$ = 0.034). On the contrary, hybrid seeds from *W. pedicellata* sired with pollen from *W. subsecunda* did not germinate as well (31.9% *vs.* 92.5% for intraspecific crosses) (Table 2) and represented a moderate isolation barrier ($RI_V$ = 0.487). Hybrid seeds from *W. subsecunda* and *W. ampla* (as pollen donor) had lower viability compared to seeds from intraspecific crosses of the latter species (73.1 *vs.* 98.8%; Table 2). This loss of viability represented a relatively weak isolation barrier ($RI_V$ = 0.149).

## DISCUSSION

Using standardized metrics or reproductive isolation indices (RI), this study presents novel information on the strength and importance of several pre- and postpollination barriers among sympatric species in the Bromeliaceae family. Previous studies on this plant group have only analyzed separate isolation barriers without employing comparative metrics. For instance, *Wendt et al. (2008)* studied prepollination barriers and suggested that they were ineffective at preventing interspecific gene flow, while *Souza et al. (2017)* found that postpollination barriers related to interspecific incompatibility led to different levels of reproductive isolation. Our research shows that reproductive isolation between species was complete and that prepollination mechanisms were more relevant as reproductive barriers between chiropterophilous bromeliads of the genus *Werauhia* from subfamily Tillandsioideae.

### The role of temporal barriers

Research has shown that flowering time can have varying effects on plant reproductive isolation, ranging from minimal to significant (mean RI = 0.375, *Christie, Fraser & Lowry, 2022*). Our study found that differences in flowering time were an important barrier to gene flow among the *Werauhia* species studied, with a mean $RI_F$ of 0.677. The overall strength of this barrier varied between years from 0.624 (2020–2021 season) to 0.712 (2012–2013 season), though indicating rather modest variation. Between species pairs, however, variation in isolation strength of flowering time ranged from as low as 0.128 to nearly complete isolation at 0.991, reflecting the diversity of flowering patterns at the study site. Similarly, the strength of this barrier varied among years between some species pairs (Table S1). This is the result of interannual variation in flowering patterns, which is primarily attributed to alterations in local climate or larger meteorological events that affect plant flowering (*McNeilly & Antonovics, 1968*; *Frankie, Baker & Opler, 1974*; *Marquis, 1988*; *Elzinga et al., 2007*). Thus, the importance of considering several flowering episodes to obtain a better RI estimation since year-to-year fluctuations may alter the magnitude of this reproductive barrier.

Our results showed that over half of the species-pair comparisons exhibited nearly complete reproductive isolation attributed to non-overlapping flowering phenology. Conversely, the remaining comparisons demonstrated either weak or moderate isolation ($RI_F$ = 0.13–0.56). However, in those instances of incomplete isolation due to temporal reproductive overlap, isolation was subsequently enhanced by a more effective premating barrier linked to floral morphology (discussed further). In two species pairs (*W. nephrolepis-W. pedicellata* and *W. nephrolepis-W. subsecunda*), we found almost complete reproductive isolation by non-overlapping phenology, while the floral morphological barriers were also significant yet redundant.

In a guild of sympatric bat-pollinated bromeliads from the genera *Pitcairnia*, *Pseudalcantarea*, and *Werauhia* in southern Mexico, *Aguilar-Rodríguez et al. (2019)* documented non-overlapping phenologies which apparently prevented interspecific pollen transfer. However, for closely related species growing in sympatry, the flowering time may be constraint by shared ancestry (*Rathcke & Lacey, 1985*). In our case, three species of

*Werauhia* bloomed during the dry season and one in the rainy period, while other sympatric species from the study site, *W. notata* and *W. haberi* (*Cascante-Marín, Trejos & Alvarado, 2017*; *Cascante-Marín, Trejos & Morales, 2019*), also flowered during the rainy season. The observed variation in flowering patterns suggests an absence of phylogenetic constraint regarding reproductive timing and suggests phenotypic plasticity that may contribute to mitigate reproductive interference.

## Mechanical barriers related to flower morphology

Differences in flower architecture can also prevent gene flow by influencing how pollen is deposited on the pollinator's body (*Grant, 1994*). Reproductive isolation through morphological differences in flower size plays a major role among some sympatric neotropical groups (*Kay, 2006*; *Ramírez-Aguirre et al., 2019*; *Albuquerque-Lima, Lopes & Machado, 2024*). In our study, species conformed into two groups: large (*W. ampla* and *W. nephrolepis*) and small-flowered species (*W. subsecunda* and *W. pedicellata*). The size disparity was nearly 2- to 3-fold between both groups, suggesting that flower-visiting bats would contact the anthers and stigma on different areas of their bodies, thereby preventing interspecific pollen flow. The pollen of the large-flowered Werauhias is likely deposited on the bat's head, while for small-flowered species, it is carried on the face (forehead and cheeks) (Fig. 4).

For species exhibiting comparable floral dimensions and overlapping phenology, such as *W. pedicellata* and *W. subsecunda*, differences in the positioning of anthers and stigma in relation to the corolla aperture are key in preventing interspecific pollen transfer. The lateral positioning of the anthers in *W. subsecunda* flowers likely causes pollen to be deposited on the bat's cheeks, while in *W. pedicellata*, pollen is probably deposited on the top region of the snout and forehead. In a previous work on the reproductive systems of the studied species (*Núñez-Hidalgo & Cascante-Marín, 2024*), we recovered pollen from *Werauhia* found on the head and snout of captured bats, but we could not identify the species it originated from due to the morphological similarities of the pollen grains.

Research has shown that in taxonomically unrelated plants pollinated by bats, the differential placement of pollen on the pollinator's body serves as an effective reproductive barrier to interspecific pollination (*Tschapka, Dressler & von Helversen, 2006*; *Muchhala & Potts, 2007*; *Muchhala, 2008*; *Stewart & Dudash, 2016*). Recently, *Pontes, Machado & Domingos-Melo (2024)* illustrated how various strategies of pollen deposition on the pollinator's body contribute to facilitating the reproductive coexistence of a guild of chiropterophilous plants in a Neotropical dry forest. *Muchhala (2008)* proposed that the evolution of this isolation mechanism is particularly facilitated in bat-pollinated plants, attributed to the larger size of bats relative to other pollinator groups, enabling more accurate pollen deposition on specific areas of their bodies. Similar studies on hummingbird-pollinated bromeliads could determine whether reproductive isolation involving pollen-placement strategies are also significant in this pollination guild or are exclusive to bat flowers.

Our estimations of the strength of mechanical barriers due to floral morphology ($RI_{MS}$ and $RI_{MP}$) were based on clear-cut and statistically significant differences in size of corolla

and reproductive organs (Fig. 3). Even though the differences in the position of stamens and stigma were sufficiently intuitive to assume the existence of a full impediment to interspecific pollen transfer, both measurements should be interpreted as conservative estimates. The high similarity in pollen grain morphology and size overlap among the studied species (unpublished data) precluded any analysis based on detecting differential pollen deposition on stigmas because of unreliable identification. Further experimental studies of specific pollen deposition on the pollinator body using pollen dyes or novel techniques (*Minnaar & Anderson, 2019*) may corroborate our interpretation.

## Interspecific incompatibility and hybrid progeny

The examined postpollination barriers showed a lower strength compared to prepollination barriers, with an average RI value of 0.225 *vs.* 0.614, respectively. The results of the reciprocal crosses indicated that interspecific incompatibility plays an inconsistent role in preventing gene flow among species pairs. In two of the three possible reciprocal crosses, the absence of fruit production reflected a marked incompatibility or incongruity (*Vervaeke et al., 2001*), although it was not symmetrical in all cases. For example, unilateral incompatibility (*Lewis & Crowe, 1958*) was observed in crosses between *W. subsecunda* and *W. ampla*, when the former acted as a pollen recipient. However, reciprocal crosses between *W. subsecunda* and *W. pedicellata* revealed partial incompatibility, which allowed the production of hybrid progeny. These permeable postpollination barriers have also been documented among bromeliad species of the genera *Aechmea* (Bromelioideae), *Pitcairnia* (Pitcairnioideae) and *Vriesea* (Tillandsioideae) (*Parton et al., 2001*; *Wendt et al., 2002*; *Souza et al., 2017*). The precise location and mechanisms of operation of this barrier are unknown.

*Covas & Schnack (1945)* explained the phenomenon of unilateral incompatibility by proposing a positive relationship between pollen size and pistil length in the two species. *Stroo (2000)*, in a compilation of studies on bat-pollinated plants, found a positive correlation between pollen size and stigma length. They suggested that the pollen grain needs to accumulate sufficient resources for tube growth as it traverses the stigma to reach the egg cell (*Levin, 1971*; *Cruden, 2009*). Moreover, the size and depth of the stigma also play a role, as the pollen grain can draw resources for tube growth from the stylar liquid (*Cruden, 2009*; *Wang et al., 2016*). In this context, the larger pollen of *W. ampla* (62–75 µm) successfully reached the ovary of *W. subsecunda*, which has a shorter pistil. Conversely, the smaller pollen of *W. subsecunda* (50–64 µm) was apparently unable to traverse the longer pistil of *W. ampla*. Although this needs confirmation through an analysis of pollen tube growth, this pattern of unilateral incompatibility has also been observed in crosses between congeneric bromeliads of the genera *Aechmea*, *Alcantarea*, and *Vriesea* (*Vervaeke et al., 2001*; *Matallana et al., 2016*; *Souza et al., 2017*).

Further isolation barriers related to seed vigor also showed varying levels of effectiveness in preventing gene flow. These levels ranged from low to moderate; however, similar to interspecific incompatibility, they were also redundant.
### Are there other potential barriers to gene flow among the studied *Werauhia*?

Although we did not directly address the following parameters in this experiment, we recognize that they may contribute to the reproductive isolation of the studied species, and we discuss them below. Variations in microhabitat specialization could serve as a spatial reproductive barrier among sympatric species (*Schluter, 2001*); however, we lack quantitative data regarding the spatial distribution or microhabitat preferences of the *Werauhia* species in question. Based on our field observations, the examined species tend to occupy similar microhabitats, typically colonizing the lower sections of tree trunks and branches within the inner areas of tree crowns, thus the role of this barrier is likely minor or null.

A temporal barrier related to the time of flower anthesis may constitute an additional mechanism of isolation (*Levin, 1971*). In the studied species, flowers open during the same period in the late afternoon (between 16–17 h) and before the nocturnal pollinator is active (*Núñez-Hidalgo & Cascante-Marín, 2024*), thus it does not constitute an isolation mechanism. Autogamy or the ability to spontaneously self-fertilize (*i.e.*, selfing) has been proposed by *Levin (1971)* as a reproductive barrier. For the Bromeliaceae family, *Matallana et al. (2016)* suggested that selfing was a mechanism to avoid hybridization, due to the high frequency of self-compatibility and autogamy among bromeliads. In a previous study, we found that our studied species showed high levels of autonomous self-fertilization (*Núñez-Hidalgo & Cascante-Marín, 2024*). This study demonstrated that selfing occurs at the end of the flower's life (*i.e.*, delayed selfing) after the opportunities for cross-pollination have diminished, primarily serving as a reproductive assurance mechanism. Additionally, differences in the number of chromosomes or ploidy levels between species may represent a postpollination barrier to prevent the formation of hybrid progeny (*Stebbins, 1950*; *Grant, 1981*). Polyploidy has been reported in several groups of bromeliads (*McWilliams, 1974*; *Brown & Gilmartin, 1983*, *1986*; *Gitaí, Horres & Benko-Iseppon, 2005*) but basic information on chromosome numbers is lacking for *Werauhia* species in general.

## CONCLUSIONS

The most significant contribution to total reproductive isolation came from prepollination barriers, which were on average 2.5 times stronger than postpollination barriers (mean RI = 0.614 *vs*. 0.225, respectively). For half of the species-paired comparisons, non-overlapping flowering schedules alone provided sufficient isolation strength to prevent gene flow ($RI_F$ values > 0.95). When flowering time was insufficient, then differences in floral size and position of reproductive organs in the flower worked in combination to establish a complete reproductive barrier. As a result, the estimates of total reproductive isolation across species pairs were complete (TI = 0.984–1.0; Table 2), suggesting the absence of gene flow between the four *Werauhia* species studied. When present, postpollination barriers were redundant and more variable in their strength. Most reproductive barriers were nearly symmetric, which means they exerted comparable strength in both directions between species pairs, except for interspecific compatibility

between *W. ampla* and *W. subsecunda*. Our results agree with the general trend described by *Christie, Fraser & Lowry (2022)* regarding the importance of prepollination barriers but contradict previous suggestions that, in the Bromeliaceae family, prepollination reproductive barriers are weak (*Wendt et al., 2008*; *Matallana et al., 2016*). Further research involving species pollinated by hummingbirds and bees will enhance our understanding of the reproductive barriers that maintain the local coexistence of highly diverse bromeliad communities. We encourage the use of reproductive isolation indices (RI) to estimate the strength and contribution of the different barriers.

## ACKNOWLEDGEMENTS

AI-powered writing tools (QuillBot) were used to enhance the grammar in some ideas.

### Funding

This work was funded by the Vice rectory of Research from University of Costa Rica (Projects C0060 and C3015). The funders had no role in study design, data collection and analysis, decision to publish, or preparation of the manuscript.

### Grant Disclosures

The following grant information was disclosed by the authors:
University of Costa Rica: C0060, C3015.

### Competing Interests

The authors declare that they have no competing interests.

### Author Contributions

- Stephanie Núñez-Hidalgo conceived and designed the experiments, performed the experiments, analyzed the data, prepared figures and/or tables, authored or reviewed drafts of the article, and approved the final draft.
- Alfredo Cascante-Marín conceived and designed the experiments, performed the experiments, prepared figures and/or tables, authored or reviewed drafts of the article, and approved the final draft.

### Data Availability

The raw measurements and calculations of reproductive isolation indices (RI) are available in the Supplemental File.

### Supplemental Information

Supplemental information for this article can be found online at http://dx.doi.org/10.7717/peerj.19652#supplemental-information.

Peer J

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
