# Peer review of "Prepollination barriers prevent gene flow between co-occurring bat-pollinated bromeliads in a montane forest"

_PeerJ, doi:10.7717/peerj.19652_

## Round 0.1 · original submission · Major Revisions

· Academic Editor

Major Revisions

Dear authors, I ask you to carefully correct the manuscript and report on each of the reviewers' fundamental comments. I hope that the new version of this manuscript will be accepted for publication.

·

Basic reporting

The article is well written, with clear and well-structured English.

However, throughout the document there is an inconsistency in the terminology they use, which makes it difficult to read and understand the article. So, I strongly recommend unifying the language.
The references are adequate and sufficient to support their arguments.
The article is well structured, with sections suitable for this type of article.

Experimental design

Although the methodology is very clear in its mesh, it is necessary to detail how the IR of the postzygotic barriers were calculated.

Validity of the findings

The supplement's Excel sheet, which is a great resource, shows the same language inconsistencies. I suggest naming each sheet in terms of the IR you are evaluating. In the case of fruit and seeds sheet, I suggest separating the data, since the number of fruits was used to evaluate RII while the number of seeds was used to evaluate the RIS. It would also help a lot if the individual barrier sheets (interspecific crosses, number of seeds and viability of hybrids) were to put on each of them the table of results of the corresponding IR calculations. Just as they did for the Phenology IR and the Mechanical IR. And that these tables remain as they are on the individual barriers sheet, so readers will have a clear idea of where they came from and how they were calculated from the original data.

Additional comments

In the discussion, they use the term Syntopic, which refers to the non-interference coexistence between species. However, it may not be entirely true in nutritional terms among these four species of tank-type bromeliads. I do not know their vertical distribution, but if they do vary, species in higher strata could most likely change both the quantity and quality of runoff that species in lower strata would receive. Likewise, it is known that some sympatric species of epiphytic bromeliads that share a pollinator (hummingbirds), move their flowering periods to retain their pollinators, which could perhaps be the case for these species. I think it's risky to assert that there is no interference of any kind between these morphologically similar species.
Also within the discussion, lines 420-424, I recommend changing the arguments given for the disadvantage of late germination (pathogens and competition), since within epiphytism these factors have little impact.

·

Basic reporting

The data on the measurements of the flower parts used for the PCA was not provided for further examination.

Experimental design

The description for Principal Component Analysis requires more details to explain how the analysis was carried out on the floral parts measurements.
among tests that should be performed for the PCA include heteroscedacity, via levene's test to ascertain that heteroscedacity is not influencing the final outcome of the PCA analysis.

Validity of the findings

The study is well written with clear concise explanation for each findings/observation.
Although evidence on the positioning of the pollen grains on the bat species was not shown in this study, it would be suffice for the author to highlight any literature regarding this in the manuscript. If there was data collected on this, it would be interesting to highlight whether the positioning of these pollen on the bat's body is consistent, and if there are some pollen that were not on the position stated, the degree of inconsistency. This will further support the premating and postmating barriers that the authors have described in the manuscript.

Additional comments

The PCA results section can be enhanced by including a table of descriptive statistics of the flower parts measurements

Reviewer 3 ·

Basic reporting

No comment

Experimental design

No comment

Validity of the findings

No comment

Additional comments

Dears
I am very pleased to send you my comments and suggestions on the paper Premating barriers prevent gene flow between cooccurring bat-pollinated bromeliads in a montane forest. The study was very well conducted and brings interesting aspects about the role of reproductive barriers in the reproductive isolation of four bromeliad species with nocturnal pollination. It is important to emphasize that these data are even more relevant when considering that we are dealing with plants pollinated by bats, an area where more reproductive isolation studies need to be conducted. In general, I feel that the authors present the context of the study well, present their results well, and treat their results very carefully and discuss them satisfactorily. In my review, I highlight small things that can be considered when reviewing the manuscript and suggest some literature that can complement the discussion of the results. Finally, I emphasize the importance of this study to the field of bat pollination ecology. Introduction

Line 67. closely related species written in continuous form on the same line can, in my opinion, be replaced to make the reading more fluent;

The authors contextualize the questions about pre-zygotic and post-zygotic barriers to isolation. In their study, Baack et al. 2015 suggest using the terms pre-pollination and post-pollination as alternatives for the types of isolation barriers. With this in mind, I believe that the authors could analyze the suggestion of Baack et al. 2015 and use it in their paper. I believe that this can add an interesting conceptual update to the text of the manuscript. Indeed, in line 149, the authors mention the term barrier before pollination.

Materials and methods

It is essential that the authors describe in detail the distribution of individuals of the species in the area studied and not only the distribution of the species in the area. Is there an estimate of the number of individuals? What is the minimum distance between the four species studied? Are there other species of the genus in the area apart from the four species studied?

Considering the previous comment, would it be possible to add an estimate of the microhabitat barrier strength with the data on the distribution of individuals? Geographic space is an important barrier that precedes the other barriers and I believe it can contribute positively to this estimate.

Throughout the manuscript, the term mouth of the corolla is used. I believe this term could be replaced by corolla aperture.

Discussion

When the authors discuss the importance of differences in the deposition of pollen grains on the bodies of bats, I think they can insert and discuss the study by Pontes et al. 2024. In this study, the authors talk about the importance of differences in pollen deposition on the bodies of bats in a community of plants pollinated by bats.
PONTES, Cristina Adriane de Souza; MACHADO, Isabel Cristina; DOMINGOS-MELO, Arthur. Floral morphology and pollen placement strategies of bat-pollinated flowers: a comparative analysis within a guild of chiropterophilous plants in a Neotropical dry forest. Revista Chilena de Historia Natural, v. 97, n. 1, p. 11, 2024.

If the authors do not add spatial isolation metrics (microhabitat), as suggested, I believe this should be mentioned in the discussion section where other barriers that may influence the reproductive isolation of bromeliad species are presented

Line 450 Werauhia must be italicized

In Figure 3b, instead of drawings of the flowers of each species, I suggest that the authors use a photograph of the flower as in Figure 4.

Finally, I would like to emphasize the beauty of the images used to illustrate the species studied, as well as the figures related to the data of the results.

---

## Round 0.2 · accepted · Accept

· Academic Editor

Accept

Dear Dr. Cascante-Marín, I am pleased to inform you that your article has been accepted for publication in our journal. I wish you the same interesting research in the future and will look forward to your next publications.

For instance: L 498 "IA tools" should be "AI tools"

·

Basic reporting

no comment

Experimental design

no comment

Validity of the findings

no comment

Additional comments

no comment